# Research on Fast and Precise Positioning Strategy of an Ultrasonic Motor Based on the Ultrasonic Friction Reduction Theory

**DOI:** 10.3390/mi13091542

**Published:** 2022-09-17

**Authors:** Weijun Zeng, Song Pan, Lei Chen, Weihao Ren, Yongjie Huan, Yongjin Liang

**Affiliations:** State Key Laboratory of Mechanics Control of Mechanical Structure of Nanjing University of Aeronautics and Astronautics, Nanjing 210016, China

**Keywords:** precision positioning, ultrasonic motor, displacement reservation value, positioning time

## Abstract

To address the problems of the large positioning error and long positioning time of the traditional positioning strategy, namely, the two-phase simultaneous power-off method (TPSPM), a new positioning strategy, called the first single-phase then two-phase power-off method (FSPTTPPM), based on the ultrasonic friction reduction theory, has been proposed in this work. This method realizes zero sliding displacement between the friction material and the stator during the torsional oscillation of the shaft by controlling the driving circle frequency and the duration of the single-phase power-off period, which reduces the deviation of the displacement reservation value. In order to verify the correctness of the driving mechanism, a test platform has been built, and two positioning strategies have been used for experimental verification. The following experimental results have been obtained: compared to TPSPM, FSPTTPPM has the advantages of higher positioning accuracy and short positioning time. In terms of the positioning accuracy, the relative errors of the displacement reservation values of FSPTTPPM and TPSPM vary with the initial angular velocity (0.24 to 1.18 rad/s) in the range of −0.4 to 0.1 and −0.8 to 0.8, respectively. In addition, the relative error of the displacement reservation value is closer to zero than that of TPSPM at the same initial angular velocity. In terms of the positioning time, when the initial angular velocity is greater than 0.7 rad/s, the positioning time of the FSPTTPPM is approximately 10 ms smaller than that of the TPSPM.

## 1. Introduction

With the development in the field of nanofabrication, the precision table technology is imposing increasingly higher requirements for the stroke, speed, and accuracy of precision positioning systems, with the control accuracy required to be in the micrometer or even nanometer range [1,2,3]. With their advantages of simple structure, fast response, and high positioning accuracy, ultrasonic motors meet the requirements of precision positioning [4,5,6] and have a wide range of applications in precision instruments, aerospace, robotics, and biomedicine [7,8,9,10]. The common positioning strategy in engineering is TPSPM, and based on this strategy, many scholars have adopted various control methods for positioning. Gencer constructed an electrical model of an ultrasonic motor using the MATLAB Simulink environment and used the driving frequency, phase difference, and duty cycle of the motor as the input quantities for proportional-integral-derivative (PID) control studies. In the position control process, the experimental data has demonstrated that the phase difference can be used as the control variable for positioning the control in the low-speed stage. However, a problem of low positioning accuracy was encountered [11,12]. Bal used a fuzzy control for the position study, taking the angular error and rotation angle frequency as the input and drive frequency as the output. The results thus obtained demonstrated that although the ultrasonic motor was able to track the reference positions for all of the ramp responses quickly [13], a large difference between the reference position and the actual position was observed. Senjyu studied the difference between the output of the reference model and the actual output and adjusted the parameters of the controller to compensate for the parameter changes of the controlled object to achieve motor position control. In addition, they combined fuzzy inference with model-referenced adaptive control to study the position control of the motor [14]. The results thus obtained demonstrated that the rotor position had good agreement with the ideal trajectory, and although it could achieve fast positioning, there was a large positioning error. The above analysis demonstrates that although a variety of control methods can be used for positioning based on TPSPM, there is a large positioning error, which indicates that it is not caused by the control algorithm but by the defects in the positioning strategy.

To solve the problem of the large relative position error, some researchers have proposed a high precision positioning strategy, namely, the microstepping positioning strategy [15], which greatly reduces the positioning error and lays a good foundation for expanding the application of traveling-wave ultrasonic motors. Based on this positioning strategy, many researchers have also adopted various control methods. Shi chose the motor driving voltage as the control variable and conducted experiments at different initial angular velocities. The experimental results demonstrated that when the motor speed is 10 µm/s, the maximum velocity error and coefficient of variation at a steady state are relatively small, but the positioning time reaches at least 20 ms. Chen achieved high precision control of the motor position from continuous motion and stepping motion using the segmental approximation strategy [16]. By choosing a specific number of pulses for multiple sets of experiments, the experimental displacement plots demonstrated that the accurate stepping resolution in both directions could reach 3.3 µrad, and this method achieved high precision localization resolution. In addition, a positioning time of at least 30 ms was taken for each set of experiments. Wang achieved control of the motor speed and displacement by controlling the number of driving waves, driving voltage, pre-pressure, and drive frequency [17]. Their experiments demonstrated that the step distance increases with the increase in the number of sinusoidal signals, which coincides with the transient analysis results. The proposed motor can output a microstep distance of approximately 0.26 µm when the number of the sinusoidal signals is 1; thus, achieving a high accuracy positioning resolution but a long positioning time of at least 30 ms is taken for each set of experiments. Shi proposed a closed-loop control strategy by using both the step control and the fuzzy PID control [18], and the controller was constructed with the function of providing a closed-loop control of the speed by adjusting the driving voltage amplitude in the stepping driving mode. Comprehensive experiments on the developed control strategy were conducted under different target speeds. There was a maximum of 24.5% speed error at the target speed of 10 μm/s, meanwhile, the coefficient of variation and the response time were 16.3% and 0.11 s, respectively. Snitka proposed the concept of a linear ultrasonic motor drive capable of nanometric steps, long-range travel and reversible controlled motion, and the motor concept developed is based on the superposition of a longitudinal and bending vibrations of a rectangular resonator [19]. The open loop positioning system with a designed stepper ultrasonic drive produced 10 nm resolution and 5% displacement repeatability. The system with computer-controlled position feedback has demonstrated 0.3 mm positioning accuracy over the 100 mm positioning range. In summary, the microstepping positioning strategy uses the transient characteristics of the ultrasonic motor and the stepping characteristics for positioning control.

In addition, Delibas proposed a new driving method for resonance drive type piezoelectric motors [20], in which the piezoelectric vibrator was excited using two driving sources at two different frequencies, and the difference between the two excitation frequencies was synchronized to the servo sampling frequency of the digital control unit. The performance of the proposed driving method was compared with those of the conventional driving methods, and it was obtained that the positioning error for the linear movements between the desired and actual positions decreased to less than 10 nm for velocities ranging from 1 mm/s to 0.001 mm/s. Giraud proposed a position-control scheme of an inertial load [21], and the guideline used for this control was a rotation of 90 degrees in a response time of about 200 ms with a position error of 0.6 mrad, targeting a typical application for avionics. Although this method has a high positioning accuracy, the problem of long positioning time exists. This method cannot provide fast positioning, and thus, in the following will not be discussed in detail.

The present study combines the advantages of the above two positioning strategies and proposes a new positioning strategy, namely, the first single-phase then two-phase power-off method (FSPTTPPM), which can achieve fast positioning and ensure small positioning errors. This paper has been divided into five sections: An introduction to the driving mechanism of the TPSPM and a detailed description of the motion characteristics of the particle of the rotor has been given in Section 2. The different torsion angle expressions, obtained on the basis of the relationship between the torque of the shaft and the maximum static friction torque, have also been described in this section. The advantages and disadvantages of the TPSPM have been summarized once again, and optimization methods have been proposed to address the shortcomings of this positioning strategy. In Section 3, a new positioning strategy, namely, the FSPTTPPM, has been proposed based on the principle of ultrasonic friction reduction, and its driving mechanism has been analyzed. In Section 4, a description of the test platform established in this work has been given, and the experimental results obtained using this platform have been presented. A comparison of the experimental results verified the correctness of the theoretical analysis. Conclusions from this study have been given in Section 5.

## 2. Characteristics of TPSPM

TPSPM is widely used in engineering applications. This method benefits from the advantage of fast response and high braking force of an ultrasonic motor power-off self-locking [22]. However, when using this method for positioning, a large torsional vibration of the shaft and sliding motion between the stator and the friction material occurs, resulting in poor positioning accuracy and long positioning time. The following section describes the characteristics of this driving method first and then analyzes the reasons for the abovementioned drawbacks.

### 2.1. Assembly Structure System of the Motor and Definition of Particles

As shown in Figure 1, one end of the rotor is connected to the rotor of the motor, and the hollow-type encoder is set on the outer side of the other end of the rotor. A particle, Q, is set on the surface of the friction material on the rotor, and a particle, W, is set on the shaft.

According to material mechanics, there are differences in the rotational speed and displacement at the two ends of the rotor because of the elastic element of the rotor. The rotational angles of the shaft and the rotor are set as $\gamma_{ro}\left(t\right)$ and $\gamma_{st}\left(t\right)$, respectively, and the theoretical position value is set as $\gamma_{zw}$, as shown in Figure 1. The rotational angular velocities of the shaft and the rotor are $\omega_{ro}\left(t\right)$ and $\omega_{st}\left(t\right)$, respectively, as shown in Figure 2.

### 2.2. Introduction to TPSPM

The driving mode of the TPSPM is shown in Figure 3. The two driving ports, sin-phase and cos-phase, are connected to the polarization regions of phase A and phase B, respectively, in the stator. The signal-driving time of the TPSPM is divided into two stages: the driving period ($t_{on}$) and the stopping period ($t_{off}$). In the driving period, the two driving ports simultaneously input sinusoidal waves with a 90° phase difference for driving the ultrasonic motor. In the stopping period, the two driving ports simultaneously stop inputting the sine wave signal.

### 2.3. Analysis of the Driving Mechanism of the TPSPM

The displacement and angular velocity of the axis obtained from the positioning experiment with the TPSPM are shown in Figure 4. The initial angular velocity of the shaft is maintained at $\omega_{ro}\left(t_{2}\right)$ when the signal-driving time is in the driving period ($t_{on}$) and a switch from the driving period to the stopping period occurs when the two-phase is powered off at the same time. The stopping period ($t_{off}$) is divided into two periods, namely, the deceleration period $\left(t_{d}\right)$ and the attenuated resonance period $\left(t_{u}\right)$. As shown in Figure 4a, the angular velocity of the rotor decreases sharply from $t_{2}$ under the action of the friction torque and the shaft begins to deform and generate the torsion angle, $\gamma_{ro}\left(t\right)$, and torsion torque, $T_{ro}$, around the centerline. The angular velocity of the rotor drops to zero at $t_{3}$, the rotation angle of the rotor shaft changes from $\gamma_{ro}\left(t_{2}\right)$ to $\gamma_{ro}\left(t_{3}\right)$ at $t_{3}$, and the angular velocity decreases from $\omega_{ro}\left(t_{2}\right)$ to $\omega_{ro}\left(t_{3}\right)$. The shaft continues to deform from $t_{3}$ and the rotation angle changes from $\gamma_{ro}\left(t_{3}\right)$ to $\gamma_{ro}\left(t_{4}\right)$ at $t_{4}$. In addition, the angular velocity of the shaft decreases from $\omega_{ro}\left(t_{3}\right)$ to zero. This period is defined as the deceleration period.

The shaft returns to the original state from the torsional deformation from $t_{4}$ and performs a damped torsional vibration until $t_{5}$, when the vibration stops. The above period is defined as the attenuated resonance period ($t_{u}$). In addition, the rotation angle of the rotary axis changes from $\gamma_{ro}\left(t_{4}\right)$ to $\gamma_{ro}\left(t_{5}\right)$. The motion characteristics of the stopping period will be described in detail in the following section.

### 2.4. Motion Characteristics of the Particle in the Stopping Period

#### 2.4.1. Analysis of the Driving Mechanism of the Deceleration Section

As shown in Figure 5, the kinematic characteristics of the deceleration period $t_{d}$ and the attenuated resonance period $t_{u}$ are shown in Figure 5. The relative sliding occurs between the friction material and the stator, and the angular velocity of the rotor and the shaft decreases sharply in the exponential form under the action of sliding friction in the period $t_{2}\sim t_{3}$. The expression for the angular velocity derived from the equation of motion is
(1)$$\left\{ \begin{array}{l} \omega_{ro}\left(t\right)=\omega_{ro}\left(t_{2}\right)e^{-\tau_{ro}t},\left(t\in \left[t_{2},t_{3}\right]\right)\\ \omega_{st}\left(t\right)=\omega_{st}\left(t_{2}\right)e^{-\tau_{st}t},\left(t\in \left[t_{2},t_{3}\right]\right) \end{array}\right.,$$
where $\tau_{ro}=\frac{c_{ro}}{J_{ro}},\tau_{st}=\frac{c_{st}}{J_{st}}$, $c_{ro}$ and $c_{st}$ represent the damping coefficients of the shaft and the rotor, respectively, and $J_{ro}$ and $J_{st}$ represent the rotational inertia of the shaft and the rotor, respectively. The motion characteristics of the particle *W* during the time period $t_{u}$ are shown in Figure 5.

As shown in Figure 1, the encoder is installed at the periphery of the shaft in the assembly structure system of the motor in this paper, and which measures the rotation angle of the shaft and the Angular velocity over time. However, the curves of the rotation angle and angular velocity of rotor particle Q change with time, which cannot be measured directly by the encoder. If the rotor speed is measured, the encoder needs to be directly installed on the rotor to accurately measure the motor speed. In addition, considering the weight of the rotor is very light, the encoder needs to be very precise and lightweight. If the weight of the encoder is large, the weight of the rotor installed with the encoder will be far greater than its own weight, which will cause the change of the moment of inertia, and then an inaccurate data measurements. Due to the limitation of experimental conditions, in order to reflect the changing trend of rotor speed, Figure 6 only shows the schematic diagram of displacement and speed, which lays the groundwork for the theoretical analysis below.

As shown in Figure 6, the power-off position has been set to $\gamma_{\mathrm{zs}}$. Since the inertia, $J_{st}$, of the rotor is very small and much smaller than that of the shaft, i.e., $J_{ro}$, the particle Q stops rotating at $t_{3}$ and is positioned at $\gamma_{\mathrm{zw}}$, and there is no sliding displacement between the friction material and the stator. The shaft is elastic and connected to the rotor by the thin aluminum material, and the angular velocity of the particle W is $\omega_{ro}\left(t_{3}\right)$ at $t_{3}$. As shown in Figure 1, the expressions for $\gamma_{ro}\left(t\right)$, torsional shear stress, $\tau_{ro}\left(t\right)$, and the torsional torque, $T_{ro}\left(t\right)$, resulting from the deformation of the shaft, are obtained according to the mechanics of the materials [23], respectively, as follows:(2)$$\left\{ \begin{array}{l} \tau_{ro}(t)=G_{ro}\gamma_{ro}(t)\\ \gamma_{ro}(t)=\frac{T_{ro}(t)h_{ro}}{G_{ro}I_{ro}} \end{array}\right.,$$
where $h_{ro}$, $G_{ro}$, and $I_{ro}$ represent the length, shear modulus, and cross-sectional polar moment of inertia of the shaft, respectively [24], and are expressed as follows:(3)$$\left\{ \begin{array}{l} G_{ro}=\frac{E_{ro}}{2\left(1+\mu_{ro}\right)}\\ I_{ro}=\frac{\pi D_{ro}{}^{4}}{64} \end{array}\right.,$$
where $R_{ro}$, $E_{ro}$, and $\mu_{ro}$ represent the radius, Young’s modulus, and Poisson’s ratio of the shaft, respectively. The maximum static friction force, $f_{smax}$, is the critical point of the static friction and sliding friction. The expression of the sliding friction force, $f_{slid}$, $f_{smax}$, and the maximum static friction torque, $T_{romax}$, without the effect of ultrasonic friction reduction are as follows:(4)$$\left\{ \begin{array}{l} f_{slid}\leq f_{smax}=F_{c}\mu_{0}(t)\\ T_{romax}=f_{smax}R \end{array}\right.,$$
where R represent stator radius. When the torque, $T_{ro}\left(t\right)$, of the shaft is not greater than $T_{romax}$, there is no sliding between the friction material and the stator, whereas when $T_{ro}\left(t\right)$ exceeds $T_{romax}$, the friction between the stator and the friction material becomes the sliding friction, and relative sliding occurs. Both these aspects are described in detail below.

1.Dynamic analysis of $T_{ro}\left(t\right)$ < $T_{romax}$;

As shown in Figure 5a, during the period $t_{3}\;\mathrm{to}\;t_{4}$, $T_{ro}\left(t\right)$ is always less than $T_{smax}$, and there is no sliding between the friction material and the stator. The shaft still keeps rotating and undergoes torsional deformation, and the torsional angle, $\gamma_{ro}\left(t\right)$, of the particle *W* changes from $\gamma_{zw}$ to $\gamma_{z0}$. As shown in Figure 5b, the angular velocity, $\omega_{ro}\left(t\right)$, of the particle *W* decreases from $\omega_{ro}\left(t_{3}\right)$ to zero under the action of the torque of the shaft. In addition to this, the rotational kinetic energy is converted to the shear strain energy during torsional deformation. The expressions for the rotational kinetic energy, $E_{vro}$, and the shear strain energy, $E_{zro}$, are shown below:(5)$$\left\{ \begin{array}{l} E_{vro}=\frac{J_{ro}{\left(\omega_{ro}(t_{3})\right)}^{2}}{2}\\ E_{zro}=\frac{{\left(\tau_{ro}\left(t\right)\right)}^{2}}{2G_{ro}}=\frac{\gamma_{ro}{\left(t\right)}^{2}G_{ro}}{2} \end{array}\right.,$$

According to the law of conservation of energy, the expression for $\gamma_{ro}\left(t\right)$ generated by deformation is as follows:(6)$$\gamma_{ro}(t)=\sqrt{\frac{J_{ro}}{G_{ro}}}\omega_{ro}(t_{3}),$$

When $T_{ro}\left(t\right)=T_{romax}$, the torsion angle of the shaft deformation is defined as the critical torsion angle, $\gamma_{romax}$. The critical torsional shear stress, $\tau_{romax}$, and $\gamma_{romax}$ are obtained according to Equation (2) as follows:(7)$$\left\{ \begin{array}{l} \tau_{romax}=\gamma_{romax}G_{ro}\\ \gamma_{romax}=\frac{T_{romax}h_{ro}}{G_{ro}I_{ro}} \end{array}\right.,$$

The critical rotational angular velocity, $\omega_{romax}$, which produces sliding at $t_{3}$, is obtained by combining Equations (4), (6) and (7) as follows:(8)$$\omega_{romax}=\frac{RF_{c}\mu_{0}h_{ro}}{I_{ro}\sqrt{J_{ro}G_{ro}}},$$

The critical torsion angle, $\gamma_{romax}$, for generating sliding is obtained by combining Equations (4) and (7) as follows:(9)$$\gamma_{romax}=\frac{h_{ro}RF_{c}\mu_{0}}{I_{ro}G_{ro}},$$

2.Dynamic analysis of $T_{ro}\left(t\right)>$ $T_{romax}$;

As shown in Figure 7a, when $T_{ro}\left(t\right)>T_{romax}$, the frictional force changes from static friction to sliding friction and generates relative sliding during the period $t_{y}$ to $t_{4}$, and the rotation angle of the particle Q changes from $\gamma_{zw}$ to $\gamma_{zwro}$. As shown in Figure 7b, the angular velocity of the particle Q is first accelerated and then decelerated by the correlation of the torque and the static friction torque of the shaft.

The rotation angle of the shaft changes from $\gamma_{zw}$ to $\gamma_{zwro}$ in the period $t_{3}\;\mathrm{to}\;t_{y}$, as shown by the blue curve in Figure 7a. $\gamma_{ro}\left(t\right)$ of the shaft starts increasing from $t_{3}$ until $\gamma_{romax}$, which causes the torque of the shaft to be increased to the maximum static friction torque, $T_{romax}$, at the same time.

In the period $t_{y}\;\mathrm{to}\;t_{4}$, the shaft continues to twist when $T_{ro}\left(t\right)$ exceeds $T_{romax}$, which causes relative sliding between the friction material and the stator. In addition, the position of the particle W changes from $\gamma_{zw}$ to $\gamma_{zwro}$. Since the frictional force between the stator and the friction material is the sliding friction in this process, the torsion angle of the rotating shaft is kept as $\gamma_{romax}$.

The red and blue curves in Figure 8 represent the rotational angle and angular velocity curves of the stator and the friction material without and with the sliding displacement, respectively. From a comparison of the red and blue curves, it can be observed that the kinetic energy of the shaft is converted into the internal energy in the process of relative sliding, and the amplitude of the angular velocity of the shaft demonstrates a significant decay.

#### 2.4.2. Analysis of the Driving Mechanism of the Attenuated Resonant Period

As shown in Figure 5 and Figure 8, during the period $t_{u}$ ($t_{4}\;\mathrm{to}\;t_{5}$), the shaft performs damped torsional vibration. The damped free vibration of the torsional oscillation system is expressed as follows [25]:(10)$$J_{ro}\frac{d\gamma_{ro}{}^{2}\left(t\right)}{dt^{2}}+c_{ro}\frac{d\gamma_{ro}\left(t\right)}{dt}+k_{ro}\gamma_{ro}\left(t\right)=0,\left(t\in \left(t_{4},t_{5}\right]\right),$$
where $k_{ro}$ represents the stiffness coefficient of the shaft and $\gamma_{p}\left(t\right)$ represents the torsional angle of the shaft. Since the vibration of the torsional pendulum system belongs to the underdamped vibration, the following expression is obtained by solving Equation (10):(11)$$\gamma_{ro}\left(t\right)=A_{r}e^{-\xi_{ro}\omega_{nro}t}\sin\left(\omega_{dro}t+\varphi_{dro}\right),\left(t\in \left(t_{4},t_{5}\right]\right),$$
where $A_{r}$ represents the initial value of the amplitude of the torsional vibration and $\varphi_{dro}$ represents the phase angle of the torsional vibration. These parameters are expressed as follows:(12)$$\left\{ \begin{array}{l} A_{r}=\sqrt{\gamma_{0}{}^{2}+{\left(\frac{\frac{d\gamma_{ro}\left(t\right)}{dt}|{}_{t=t_{4}}+\xi_{ro}\omega_{nro}\gamma_{0}}{\omega_{dro}}\right)}^{2}}\\ \tan\varphi_{dro}=\frac{\gamma_{0}\omega_{dro}}{\xi_{ro}\omega_{nro}\gamma_{0}+\frac{d\gamma_{ro}\left(t\right)}{dt}|{}_{t=t_{4}}} \end{array}\right.,$$

The initial conditions of the torsion pendulum system in the above equation are as follows:(13)$$\left\{ \begin{array}{l} \gamma_{0}=\sqrt{\frac{J_{ro}}{G_{ro}}}\omega_{ro}(t_{4})\\ \frac{d\gamma_{ro}\left(t\right)}{dt}|{}_{t=t_{4}}=\omega_{ro}(t_{4}) \end{array}\right.,$$

In Equation (12), $\xi_{ro}$ represents the damping ratio of the shaft, $\omega_{nro}$ represents the undamped resonance frequency of the shaft, and $\omega_{dro}$ represents the damped resonance frequency of the shaft. These parameters are expressed as follows:(14)$$\left\{ \begin{array}{l} \omega_{nro}=\sqrt{\frac{k_{ro}}{J_{ro}}}=\frac{\omega_{dro}}{\sqrt{1-\xi_{ro}{}^{2}}}\\ \xi_{ro}=\frac{c_{ro}}{2\sqrt{k_{ro}J_{ro}}} \end{array}\right.,$$

The resonance period of the shaft in the process of torsional vibration is expressed as follows:(15)$$\left\{ \begin{array}{l} T_{nro}=\frac{2\pi }{\omega_{nro}},T_{dz}=\frac{2\pi }{\omega_{dro}}\\ T_{dro}=\frac{T_{nro}}{\sqrt{1-\xi_{ro}{}^{2}}} \end{array}\right.,$$
where $T_{nro}$ represents the undamped resonance period of the shaft and $T_{dro}$ represents the damped resonance period of the shaft. The entire positioning process is finished when the rotor shaft and rotor completely stop torsional vibration ($t_{5}$ in Figure 8).

### 2.5. Analysis of the Twist Angle of the Rotor

#### 2.5.1. Torsional Angle Analysis When the Torque of the Shaft Is Not Greater than $T_{\mathrm{romax}}$

As shown in Figure 6a, the angular velocity of the particle *Q* starts from $t_{2}$ and decelerates to zero at $t_{3}$. Since the rotational inertia, $J_{st}$, of the rotor is extremely small, the angular velocity of the particle *Q* varies approximately linearly with time and $t_{d}$ can be expressed as
(16)$$t_{d}=\frac{J_{st}\omega_{st}(t_{2})}{T_{zu}(t)+T_{load}},$$
where $T_{zu}\left(t\right)$ and $T_{laod}\left(t\right)$ represent the friction torque and load torque, respectively, where $T_{zu}\left(t\right)=Rf_{smax}$, and R represents the stator radius. As shown in Figure 6b, the expression for the rotation angle, $\Delta \gamma_{d}$, of the rotor during $t_{d}$ can be expressed as
(17)$$\Delta \gamma_{d}=\frac{1}{2}\omega_{st}(t_{2})t_{d}=\frac{1}{2}\frac{J_{st}{\left(\omega_{st}(t_{2})\right)}^{2}}{T_{zu}(t)+T_{load}},$$

During the time period $t_{u}$, since the shaft has a large rotational inertia, $J_{ro}$, and the torsional angle position value of the rotor remains constant, it needs to go through several cycles of torsional vibration to stop the vibration, and particles of the rotor are finally positioned at $\gamma_{zw}$. In the positioning process, the power needs to be cut off in advance before the rotor reaches the target position in order to leave the required displacement reserve value for the rotor to slow down. In this work, the displacement reserve value derived from the theoretical formula has been defined as the theoretical displacement reserve value, $\Delta \gamma_{o}$, whereas that obtained from the experiment has been defined as the measured displacement reserve value, $\Delta \gamma_{oc}$. According to Figure 6a, the expression for $\Delta \gamma_{o}$ is
(18)$$\Delta \gamma_{0}=\Delta \gamma_{d}=\frac{1}{2}\frac{J_{st}{\left(\omega_{st}(t_{2})\right)}^{2}}{T_{zu}(t)+T_{load}},$$

#### 2.5.2. Dynamic Analysis When the Torque of the Shaft Is Greater than $T_{\mathrm{romax}}$

When the torque of the shaft is greater than $T_{romax}$, the torque transmitted by the shaft to the rotor causes relative sliding between the friction material and the stator, and thus $\gamma_{ro}\left(t\right)$ of the shaft maintains the value of $\gamma_{romax}$. The expressions of the resulting sliding displacement, $\Delta \gamma_{zw}$, and the theoretical displacement reservation, $\Delta \gamma_{o}$, according to Figure 8a, are as follows:(19)$$\left\{ \begin{array}{l} \Delta \gamma_{zw}=\gamma_{zwro}-\gamma_{zw}\\ \Delta \gamma_{0}=\gamma_{romax}+\Delta \gamma_{zw} \end{array}\right.,$$

The shaft requires multiple cycles of damped decaying vibration to reach the stop position, and multiple slides occur between the friction material and the stator. Since the friction force will switch back and forth between sliding friction and static friction during this period, resulting in a relatively strong nonlinear creep between the stator and the friction material [26,27], it is difficult to find a theoretical formula that can accurately calculate the misaligned sliding displacement.

To solve this problem, a new positioning strategy, namely, FSPTTPPM, has been proposed in this study, which can ensure that the crawling between the friction material and the stator is avoided during the torsional oscillation of the shaft so that the value of the sliding displacement, $\Delta \gamma_{zw}$, tends to zero ($\Delta \gamma_{zw}\to$0). In addition, it is necessary to calculate the rotation angle, $\Delta \gamma_{off}$, of the speed reduction period to obtain an accurate displacement reservation value, $\Delta \gamma_{o}$, and avoid the search for a theoretical formula that can accurately calculate the misaligned sliding displacement, $\Delta \gamma_{zw}$. The mechanism of this positioning strategy is described in detail in the following section.

## 3. Introduction to FSPTTPPM and Analysis of Its Driving Mechanism

### 3.1. Introduction to the FSPTTPPM Driving Method

The signal-driving time of FSPTTPPM is divided into two periods, the driving period ($t_{son}$) and the stopping period ($t_{soff}$), where the stopping period is divided into the single-phase power-off period ($t_{sd}$) and the two-phase power-off period ($t_{su}$), as shown in Figure 9. During the time period $t_{son}$, the motor speed is in a steady state, and the driving circle frequency is set to $\omega_{q}$. When the signal-driving time starts to enter $t_{sd}$, the two driving ports of the motor output only one driving signal during $t_{sd}$ and the driving circle frequency is set to $\omega_{u}$. When $\omega_{ro}\left(t_{03}\right)$ is much smaller than $\omega_{romax}$, the two driving ports stop outputting the driving signals, and the signal-driving time enters the time period $t_{su}$. In addition, the shaft stops rotating at $t_{04}$. The entire positioning process is completed after the above process.

### 3.2. Principal Analysis of the FSPTTPPM

The displacement and angular velocity of the axis obtained from the positioning experiment, performed by employing FSPTTPPM, is shown in Figure 10. The angular velocity of the shaft is stabilized at the initial angular velocity $\omega_{ro}\left(t_{02}\right)$ during $t_{son}$, and the rotation angle changes from $\gamma_{ro}\left(t_{01}\right)$ to $\gamma_{ro}\left(t_{02}\right)$. When the signal-driving time is in the time period $t_{sd}$, the rotation angle changes from $\gamma_{ro}\left(t_{02}\right)$ to $\gamma_{ro}\left(t_{03}\right)$.

The control system keeps the stator and the friction material in the ultrasonic friction reduction by controlling the driving circle frequency, $\omega_{u}$ [28,29], such that the equivalent friction coefficient, $\bar{\mu_{w}}$, Appendix A is smaller than the sliding friction coefficient, $\mu_{0}$. The sliding friction force, $f_{slidc}$, and the maximum static friction force, $f_{scmax}$, in the case of ultrasonic friction reduction can be obtained by multiplying both sides of Appendix A by the pre-pressure, $F_{c}$, as follows:(20)$$f_{scmax}=f_{slidc}=F_{c}\bar{\mu_{w}}\left(t\right)<f_{smax},$$

According to Equation (20), the maximum static friction, $f_{scmax}$, generated during $t_{sd}$ in Figure 9 under the effect of ultrasonic friction reduction is smaller than the maximum static friction, $f_{smax}$, during $t_{sd}$ in Figure 3, such that the stator and the friction material are in a relative sliding state. In addition, the torsion angle and angular velocity of the shaft connected with the rotor are the critical torsion angle $\gamma_{cmax}$ and critical rotational angular velocity $\omega_{rocmax}$ under ultrasonic friction reduction, respectively. Further, the torque generated by the torsional deformation of the rotor during this time is the critical torsional torque, $T_{rocmax}$, in the case of ultrasonic friction reduction. The expressions for $\gamma_{cmax}$ and $T_{rocmax}$ are as follows:(21)$$\left\{ \begin{array}{l} \gamma_{cmax}=\frac{Rh_{co}F_{c}\bar{\mu_{w}}\left(t\right)}{G_{ro}}\\ \omega_{rocmax}=\frac{RF_{c}\bar{\mu_{w}}\left(t\right)h_{ro}}{I_{ro}\sqrt{J_{ro}G_{ro}}}\\ T_{rocmax}=\frac{I_{ro}G_{ro}\gamma_{cmax}}{h_{ro}} \end{array}\right.,$$

On combining Equations (4), (9), (20) and (21), it is found that $\gamma_{cmax}$ between the stator and the friction material with sliding during $t_{sd}$ is less than $\gamma_{romax}$ given by Equation (9). The critical torque, $T_{rocmax}$, of the shaft with the effect of ultrasonic friction reduction is less than the critical torque, $T_{pmax}$, without the effect of ultrasonic friction reduction obtained by combining Equations (4) and (21), i.e.,
(22)$$\left\{ \begin{array}{l} \gamma_{cmax}<\gamma_{romax}\\ T_{rocmax}<T_{romax} \end{array}\right.,$$

As shown in Figure 10, the stator and the friction material come in complete contact during $t_{su}$, the control system decelerates the angular velocity of the particle W from $\omega_{ro}\left(t_{02}\right)$ to $\omega_{ro}\left(t_{03}\right)$, and makes $\omega_{romax}>\omega_{ro}\left(t_{03}\right)$ during $t_{sd}$. Since the angular velocity of the particle W is much smaller than $\omega_{romax}$ after deceleration in the time period $t_{sd}$ and $\gamma_{ro}\left(t\right)$ is proportional to $\omega_{ro}\left(t_{3}\right)$ according to Equation (6), $\gamma_{ro}\left(t\right)$ is smaller than $\gamma_{romax}$. The following inequality is obtained by combining Equations (6) and (22):(23)$$\gamma_{ro}(t)=\sqrt{\frac{J_{ro}}{G_{ro}}}\omega_{p}(t_{3})<\sqrt{\frac{J_{ro}}{G_{ro}}}\omega_{romax}=\gamma_{romax},$$

Since the stator and rotor are sliding during $t_{sd}$, the torque, $T_{ro}\left(t\right)$, of the shaft is greater than $T_{rocmax}$, and the expression obtained by combining Equations (21)–(23) is as follows:(24)$$T_{rocmax}<T_{ro}<T_{romax},$$

The critical torque for the occurrence of the sliding motion between the friction material and the stator changes from $T_{rocmax}$ to $T_{romax}$ during $t_{su}$. According to Equation (24), the torque of the shaft during $t_{su}$ is less than the critical torque, $T_{romax}$, which changes the friction force between the stator and the friction material from sliding friction to static friction. Thus, the friction material and stator will not slide during the torsional vibration of the shaft.

### 3.3. Motion Characteristics of the Shaft

The blue and red curves in Figure 11 and Figure 12 represent the characteristic curves of the motion of the particle W (blue curve in Figure 7) and particle Q (blue curve in Figure 6) when the angular velocity, $\omega_{ro}\left(t_{3}\right)$, of the particle W is greater than $\omega_{romax}$ using TPSPM and FSPTTPPM, respectively.

As shown in Figure 11, when the signal-driving time is between $t_{03}\;\mathrm{to}\;t_{h}$, the rotation angle of the particle Q changes from zero to $\gamma_{zws}$ and the angular velocity of the particle W decreases from $\omega_{ro}\left(t_{02}\right)$ to $\omega_{ro}\left(t_{03}\right)$ under the action of the friction torque. In addition, as shown in Figure 12, the angular velocity of the particle W decreases from $\omega_{ro}\left(t_{02}\right)$ to zero. $t_{sd}$ can be expressed as
(25)$$t_{sd}=\frac{J_{st}\omega_{st}(t_{02})}{T_{szu}+T_{load}},$$
where $T_{szu}$ represents the resistance torque in $t_{sd}$ that can be expressed as
(26)$$T_{szu}=F_{c}\bar{\mu_{w}}\left(t\right)R,$$

The angular velocity of the particle W decreases approximately linearly. The expressions for the torsion angle during $t_{sd}$ and the theoretical displacement reservation, $\Delta \gamma_{o}$, are
(27)$$\Delta \gamma_{0}=\gamma_{zws}-\gamma_{zw}=\frac{1}{2}\omega_{p}(t_{2})t_{soff}=\frac{1}{2}\frac{J_{st}{\left(\omega_{p}(t_{2})\right)}^{2}}{T_{szu}+T_{load}},$$

For the signal-driving time during $t_{su}$, the angular velocity of the particle W decreases from $\omega_{ro}\left(t_{03}\right)$ to zero during the damped vibration, and the rotation angle changes from $\gamma_{zw}$ to $\gamma_{zws}$. Based on the above analysis, the expressions for the positioning time ($T_{dj}$) are obtained from the analysis of the driving mechanisms shown in Figure 4 and Figure 10, as follows:(28)$$T_{dj}=\left\{ \begin{array}{l} t_{d}+t_{u}\\ t_{sd}+t_{su} \end{array}\right.,$$

When TPSPM is used for positioning, the positioning time is the sum of $t_{d}$ and $t_{u}$. When FSPTTPPM is used for positioning, the positioning time is the sum of $t_{sd}$ and $t_{su}$. The total time when the amplitude of torsional oscillation of the shaft decays to less than 5 arcsec from $t_{off}$ is defined as the positioning time.

### 3.4. Parameter Setting and Analysis of the Theoretical Equations

In this study, the pre-pressure and load torque between the stator and the rotor were taken to be $F_{c}$ = 180 N and $T_{load}$ = 0.2 N·m, respectively. The parameters in the abovementioned equations have been listed in Table 1.

Firstly, the two positioning strategies are used at critical angular velocity, respectively, in the following sections; then, the parameters listed in Table 1 are substituted into the theoretical formula above; finally, the position and angular velocity are simulated with time by MATLAB, as shown in Figure 13. In the simulation, the critical angular velocity $\omega_{romax}$ and $\omega_{rocmax}$ are set as 1.35 × 10^−3^ rad/s and 0.38 × 10^−3^ rad/s, respectively, and the driving circle frequency in $t_{sd}$ is set as $\omega_{u}=2\pi \times 42,000\;\mathrm{rad}/s$, and $T_{szu}=0.7$ N·m is obtained, which is shown in Appendix A. The following three points are obtained:

When TPSPM is used for positioning, the initial angular velocity $\omega_{ro}\left(t_{3}\right)$ of the shaft in ($t_{u}$) is related to the initial angular velocity $\omega_{ro}\left(t_{2}\right)$ and the parameters listed in Table 1 after two driving ports stop outputting the driving signals; thus, $\omega_{ro}\left(t_{3}\right)$ is not controllable. When FSPTTPPM is used for positioning, the initial angular velocity $\omega_{ro}\left(t_{03}\right)$ of the shaft in ($t_{\mathrm{su}}$) is not only related to the initial angular velocity $\omega_{ro}\left(t_{02}\right)$ and the parameters listed in Table 1, moreover, it is related to driving circle frequency ($\omega_{u}$) and the duration of $t_{st}$, so $\omega_{ro}\left(t_{03}\right)$ is parameterable.

When TPSPM is used for positioning, and the initial angular velocity is large, the sliding motion between the friction material and the stator during the torsional vibration of the shaft can lead to poor positioning accuracy. If FSPTTPPM is employed for positioning, no sliding displacement occurs between the friction material and the stator regardless of the rotational inertia and the initial angular speed of the shaft, and thus the sliding displacement is zero ($\Delta \gamma_{zw}\to 0$). Hence, the search for a theoretical formula that can accurately calculate the sliding displacement is avoided.

When TPSPM is used for positioning, the angular velocity of the rotor drops sharply to zero in an exponential manner during $t_{sd}$, making $t_{d}$ very short. According to Equations (12) and (15), $A_{r}$ is positively correlated with $J_{ro}$, $T_{dro}$, and $\omega_{ro}\left(t_{3}\right)$, respectively, and $T_{dro}$ is proportional to $\sqrt{J_{ro}}$. In addition, the initial angular velocities, $\omega_{ro}\left(t_{2}\right)$ and $\omega_{ro}\left(t_{3}\right)$, are positively correlated according to Newton’s law. When the rotational inertia of the shaft and $A_{r}$ are large, the torsional vibration amplitude of the shaft needs to decay to zero after several oscillation cycles. When FSPTTPPM is used for positioning, the angular velocity of the rotor is decelerated during $t_{sd}$ based on the ultrasonic friction reduction mechanism. Although $t_{sd}$ is slightly larger than $t_{d}$, $A_{r}$ becomes small after deceleration during $t_{sd}$, making $t_{su}$ smaller than $t_{u}$. According to the definition given in Equation (28), the positioning time of FSPTTPPM is smaller than that of TPSPM.

The above three advantages were found from the theoretical analysis. To verify the correctness of the above analysis conclusions, an experimental platform was set up for experimental verification, which has been described in the following section.

## 4. Construction of the Test Platform

### 4.1. Introduction to the Test Platform

As shown in Figure 14, the test platform consists of a motor driving control system, a high-precision measuring device, a pressure measuring device, and a host computer testing system. The high-precision measuring device consists of a high-precision encoder and a shaft to measure the motor speed. The pressure measuring device consists of a pressure sensor, a torque disk, a load, a fixed vertical plate, a shaft, and mechanical connecting parts for fixing each device. The weight is pulled by the string on the outer ring of the torque disk, and the load torque is obtained by the product of the pressure measured by the pressure sensor and the rotor radius. The motor driving control system consists of the core board circuit, driving control circuit, push-pull circuit, and serial communication circuit. The host computer test system has been written by the upper computer program based on LabVIEW, which is used for sending the control instructions to the digital signal processor (DSP) and receiving data for real-time display and storage.

### 4.2. Framework of the Test System

To be able to use both TPSPM and FSPTTPPM positioning strategies for positioning, the host computer needs to send commands to the serial communication unit of the DSP28335 core board according to the communication protocol command format. After receiving the command, the control unit of the core board sends different parameters to the driving control board according to different positioning strategies, and the ultrasonic motor works under the control of the driving control board. The block diagram of the motor position test system is shown in Figure 15.

### 4.3. Structure of the Control System in the Test System

To enable the use of the two positioning strategies for performing comparative experiments, the control system of the developed test platform uses a proportional-integral-derivative (PID) closed-loop controller to control the angular speed of the motor and an open-loop method to control the angular position of the motor. The structure of the control system is shown in Figure 16.

As shown in Figure 16, the angular velocity of the motor is first controlled by a PID controller and kept in a steady state to provide different initial angular velocities for the experiments described in Section 4.4 below. Then, the power-off position value, $S_{dw}$, is set, and, finally, the two positioning strategies are used for positioning at a certain angular speed, respectively. The open-loop position resolution of the ultrasonic motor is 1 arcsec, and the sampling time is 160 µs.

### 4.4. Experiments and Analysis

#### 4.4.1. Definition of the Power-Off Reservation Value

As shown in Figure 17, $\gamma_{zc}$, $\gamma_{zs}$, and $\gamma_{zw}$ represent the measured position value, the power-off position value, and the theoretical position value, respectively. $\Delta \gamma_{o}$ and $\Delta \gamma_{oc}$ represent the reserved value of the theoretical and measured displacement, respectively. The power-off position value of $\gamma_{zs}$= 623,782 arcsec was set in the experimental process described below. Due to the deviation between the theoretical calculation and the measured position, the reserved deviation of the theoretical displacement, $\Delta \gamma_{olc}$, is defined as the difference between $\Delta s_{oc}$ and $\Delta s_{ol}$, i.e.,
(29)$$\Delta \gamma_{olc}=\Delta \gamma_{oc}-\Delta \gamma_{o},$$

In this study, the deviation error rate of the theoretical displacement reserve value has been defined to measure the positioning accuracy at different initial angular speeds, which is expressed as
(30)$$\sigma_{y}=\frac{\Delta \gamma_{olc}}{\Delta \gamma_{o}},$$

According to the above equation, the closer the value of $\sigma_{y}$ is to 0, the smaller is the value of $\Delta \gamma_{olc}$.

#### 4.4.2. Experimental Test and Analysis of the Positioning Method Based on TPSPM

Based on the reasons for the low positioning accuracy mentioned in Section 2.5.2, five sets of experiments were carried out at different initial angular velocities. Figure 18 shows the measurement chart of the position value and the angular velocity as a function of time based on TPSPM at different initial angular velocities.

Based on the values of $\Delta \gamma_{o}$ and $\Delta \gamma_{olc}$ obtained from Equations (27) and (29), respectively, and the changing trend of $T_{dj}$ and $\Delta \gamma_{oc}$ with the increasing initial angular velocity obtained by Equation (28), the results obtained from the experimental measurements are shown in Figure 19.

Figure 19 shows that $\Delta \gamma_{oc}$, $\Delta \gamma_{o}$, $T_{dj}$, and $\Delta \gamma_{olc}$ are positively correlated with the initial angular velocity, which confirms that the sliding displacement, $\Delta \gamma_{zw}$, is the reason for the large deviation of the theoretical displacement reservation value. In addition, the experimentally measured positioning time and the variation trend of Equation (16) are also correlated.

#### 4.4.3. Setting Experiment of the Driving Circle Frequency during $t_{sd}$

To achieve positioning with a short positioning time and a small deviation of $\Delta s_{olc}$, five sets of single-phase power-off positioning experiments based on FSPTTPPM and at different driving circle frequencies, $\omega_{u}$, were conducted, as shown in Figure 20.

From Figure 19, it can be observed that at the same initial angular velocity, $T_{dj}$ increases with increasing circle frequency difference between the driving circle frequency, $\omega_{u}$, and the resonance frequency (40,900 Hz). To achieve a shorter positioning time, the driving circle frequency of $t_{sd}$ is set to the driving circle frequency that is closer to the resonant frequency ($\omega_{u}=2\pi \times 41,000\;\mathrm{rad}/s$). This has been conducted based on the conclusion that the resistance torque between the stator and the friction material is positively related to the circle frequency difference obtained from the trend of the variation curve in the Appendix A.

#### 4.4.4. Experiments Employing the Two Positioning Strategies

To verify that the deviation of the theoretical displacement reservation value, $\Delta s_{olc}$, of FSPTTPPM is smaller than that of TPSPM, the superposition drive method [9] and the driving control by the PID controller were used to obtain five sets of different initial angular velocities, which are 0.24, 0.36, 0.72, 1.09, and 1.18 rad/s. Figure 21 shows the positioning value and the angular velocity obtained for different initial angular velocities when the two positioning strategies are employed.

To analyze the experimentally measured results shown above in detail, firstly, $\Delta \gamma_{oc}$ was obtained from the difference between $\gamma_{zs}$ and $\gamma_{zc}$ in Figure 20. Secondly, $\Delta \gamma_{o}$ was obtained for TPSPM and FSPTTPPM using Equations (19) and (27), respectively, and they represent the friction torque $T_{zu}$ and $T_{szu}$ in Equation (26). Equation (26) was obtained from the resistive friction torque measurement value in Appendix A. $\Delta s_{olc}$ of the two positioning strategies were obtained using Equation (29) and then $\sigma_{y}$ of the two positioning strategies was obtained using Equation (30). Finally, $T_{dj}$ values for the two positioning strategies were obtained using Equation (28). It has been observed that when the initial angular velocity changes from 0.24 to 1.18 rad/s, $\Delta \gamma_{oc}$, $\Delta \gamma_{o},\;\Delta \gamma_{olc}$, and $T_{dj}$ increase with the increasing initial angular velocity, as shown in Figure 22.

As shown in Figure 22, when the angular velocity of the driving period is less than 0.3 rad/s, the difference in the positioning time between the two positioning strategies is very small, whereas when the initial angular velocity is greater than 0.7 rad/s, the positioning time of FSPTTPPM is 10 ms, which is less than that of TPSPM. Using the experimental values plotted in Figure 21, $\sigma_{y}$ was calculated for the two positioning strategies, as shown in Figure 23. When the initial angular velocity is less than 0.44 rad/s or more than 0.73 rad/s, the error rate of the theoretical displacement reserved value based on FSPTTPPM is less than that of TPSPM. However, when the initial angular velocity is more than 0.44 rad/s and less than 0.52 rad/s, the error rate of the theoretical displacement reserved value based on FSPTTPPM is more than that of TPSPM.

#### 4.4.5. Conclusions from the Experimental Measurements Using the Two Positioning Strategies

The following three conclusions were obtained after analyzing the experimental data:
When the initial angular velocity is greater than 0.7 rad/s, the positioning time of FSPTTPPM is less than that of TPSPM.$\sigma_{y}$ of FSPTTPPM and TPSPM for the initial angular velocity from 0.24 to 1.18 rad/s varies in the ranges of −0.4 to 0.1 and −0.8 to 0.8, respectively. Compared to TPSPM, $\sigma_{y}$ of FSPTTPPM is closer to zero.When the motor is used for positioning on a project, the initial angular velocity of the most closed-loop positioning controller is quite slow. According to the variation trend of the above curve, the error rate of the theoretical displacement reserved value based on TPSPM is less than that of FSPTTPPM at low speed.

In summary, FSPTTPPM not only has a shorter positioning time but also leads to the error rate of reservation deviation to be close to zero.

## 5. Conclusions

In this study, a new positioning strategy based on the principle of ultrasonic friction reduction, namely FSPTTPPM, has been proposed, which has the following advantages compared to the traditional TPSPM strategy:
From the analysis of the experiment of the driving circle frequency setting of $t_{sd}$, it is found that $T_{dj}$ increases with increasing circle frequency difference between the driving circle frequency, $\omega_{u}$, and the resonant resonance frequency for the same initial angular velocity. A driving circle frequency set to $\omega_{u}=2\pi \times 41,000\;\mathrm{rad}/s$ in this study thus realizes positioning with a shorter positioning time.When the TPSPM strategy is used for positioning and the torque of the shaft is greater than $T_{romax}$, the torque transmitted by the shaft to the rotor causes relative sliding between the friction material and the stator, and the shaft requires several cycles of damped vibration attenuation to reach the stopping position. Due to the relatively strong nonlinear creeping between the stator and the friction material during this process, it is difficult to find a theoretical formula that can accurately calculate the misaligned sliding displacement. To solve this problem, a new positioning strategy, namely, FSPTTPPM, has been proposed in this study, which is based on the principle of ultrasonic friction reduction. It keeps the friction material and the stator in a sliding state by controlling the driving circle frequency, $\omega_{u}$, such that that no sliding occurs between the friction material and the stator during the torsional vibration of the shaft and the sliding displacement $\Delta \gamma_{zw}$ tends to zero. Thus, the search for a theoretical formula that can accurately calculate the sliding displacement is avoided, and by simply using Equation (27), an accurate displacement reservation value, $\Delta \gamma_{o}$, can be obtained.When the two positioning strategies are used for positioning, $t_{d}$ and $t_{sd}$ are almost equal, but $t_{su}$ is significantly smaller than $t_{u}$. Thus, the positioning time of FSPTTPPM is smaller than that of TPSPM. In addition, when using TPSPM for positioning, the positioning time is not only positively related to the initial angular velocity but also positively related to the rotational inertia of the shaft. However, FSPTTPPM not only has the advantage of short positioning time but also a significantly reduced influence of the rotational inertia of the shaft on the positioning.

## Figures and Tables

**Figure 1 micromachines-13-01542-f001:**
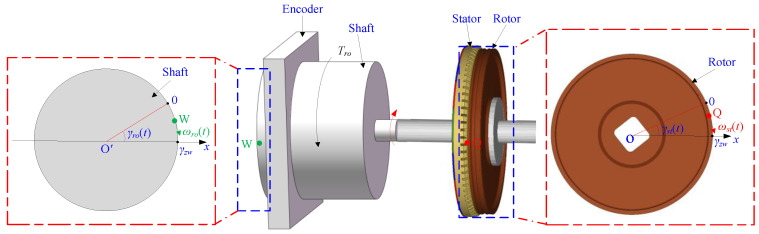
Assembly structure system of the motor and definition of particles.

**Figure 2 micromachines-13-01542-f002:**
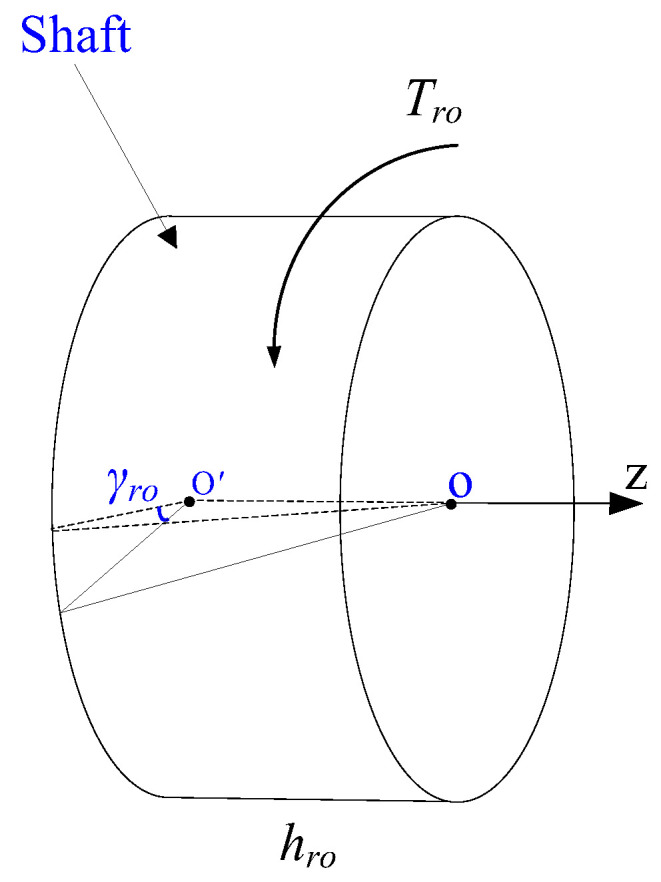
Torsional deformation diagram of the shaft.

**Figure 3 micromachines-13-01542-f003:**
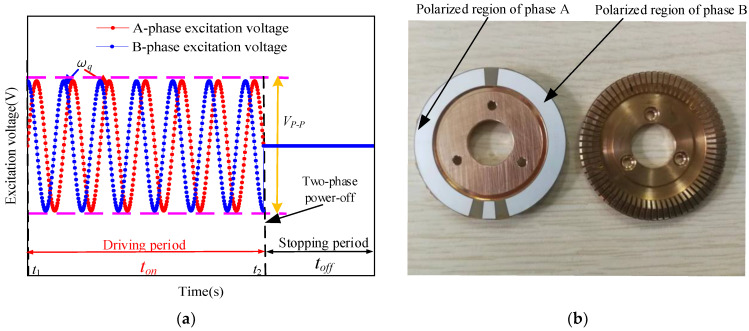
Schematic showing the working of the driving method of the TPSPM. (**a**) Shows the driving method TPSPM and (**b**) shows the polarized region of the two phases in the stator.

**Figure 4 micromachines-13-01542-f004:**
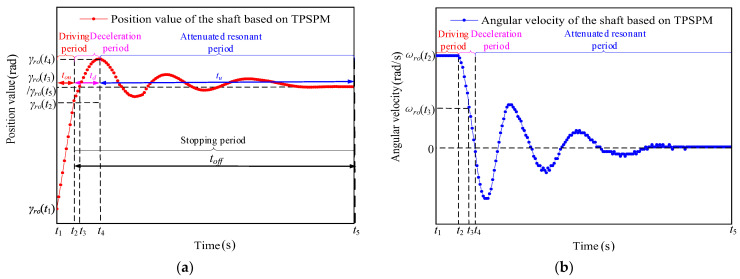
Motion characteristics of the (**a**) displacement and (**b**) angular velocity of the shaft as a function of time based on TPSPM.

**Figure 5 micromachines-13-01542-f005:**
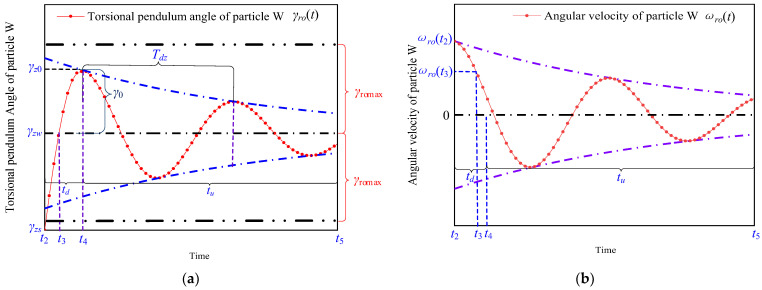
Motion characteristics of the particle W during $t_{off}$ in terms of its (**a**) rotation angle and (**b**) angular velocity as a function of time.

**Figure 6 micromachines-13-01542-f006:**
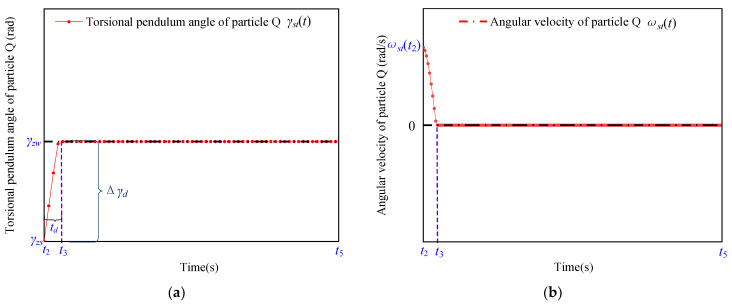
Motion characteristics of the rotor during $t_{off}$ in terms of its (**a**) rotation angle and (**b**) angular velocity as a function of time.

**Figure 7 micromachines-13-01542-f007:**
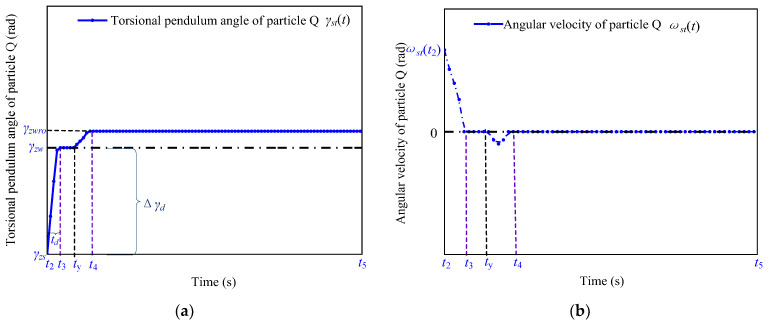
Kinematic characteristic of the particle Q during $t_{off}$ in terms of its (**a**) rotation angle and (**b**) angular velocity as a function of time.

**Figure 8 micromachines-13-01542-f008:**
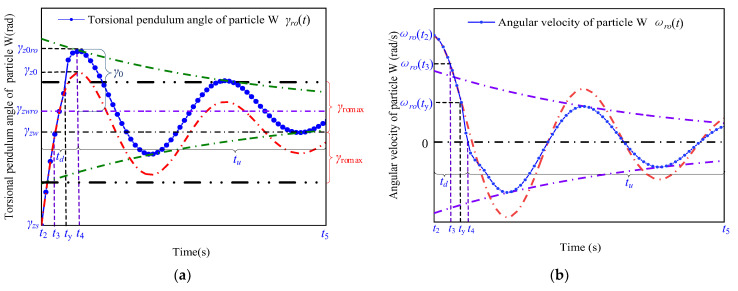
Motion characteristics of the shaft during $t_{off}$ in terms of its (**a**) rotation angle and (**b**) angular velocity as a function of time.

**Figure 9 micromachines-13-01542-f009:**
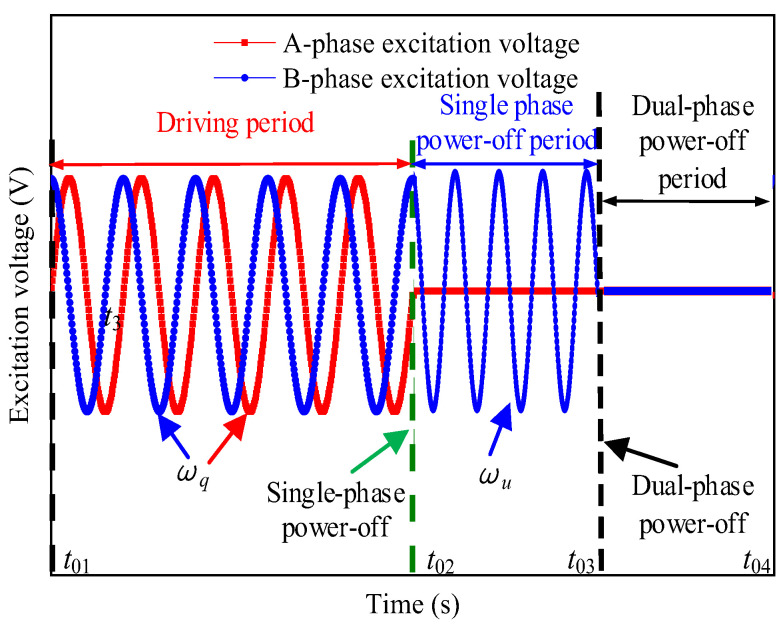
Schematic showing the driving mode of FSPTTPPM.

**Figure 10 micromachines-13-01542-f010:**
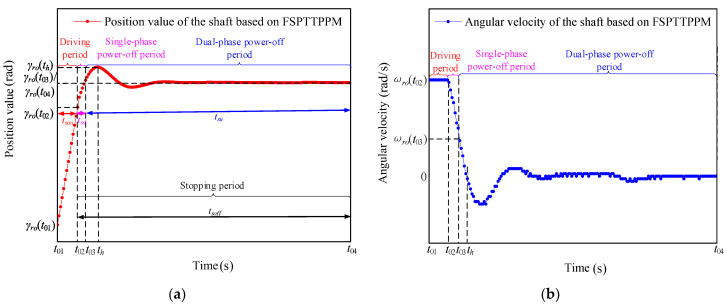
Motion characteristics of the (**a**) rotor displacement and (**b**) angular velocity of the shaft as a function of time-based on FSPTTPPM.

**Figure 11 micromachines-13-01542-f011:**
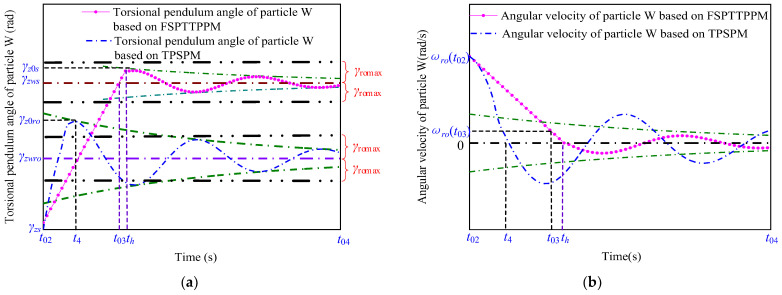
Motion characteristics of the particle W in terms of its (**a**) rotation angle and (**b**) angular velocity as a function of time.

**Figure 12 micromachines-13-01542-f012:**
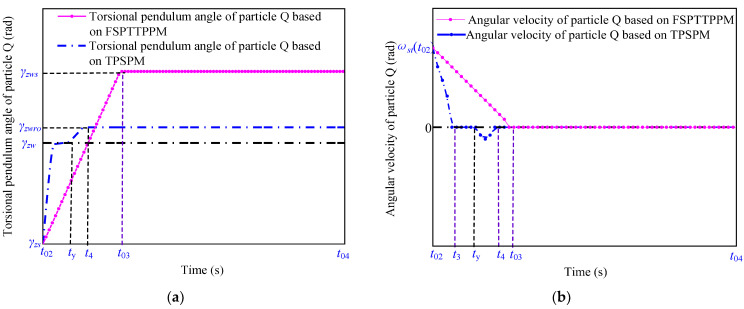
Motion characteristics of the particle Q in terms of its (**a**) rotation angle and (**b**) angular velocity as a function of time.

**Figure 13 micromachines-13-01542-f013:**
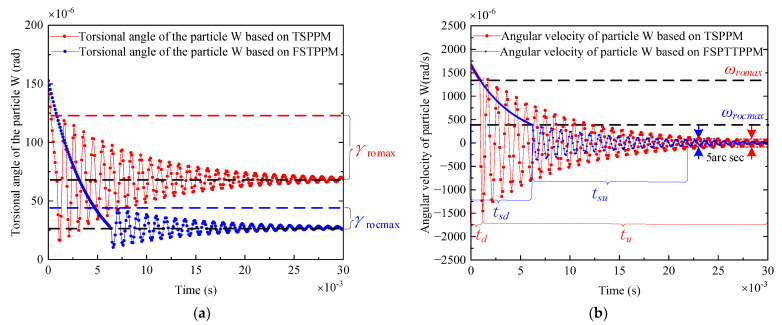
Schematic showing that the position and angular velocity are measured with time by MATLAB simulation. (**a**) the position value of the simulation and (**b**) the angular velocity of the simulation.

**Figure 14 micromachines-13-01542-f014:**
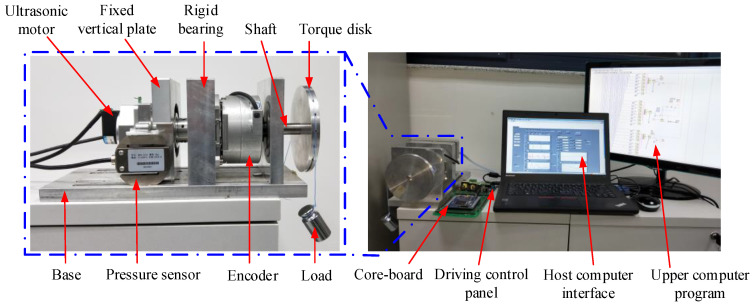
Construction of the experimental platform.

**Figure 15 micromachines-13-01542-f015:**
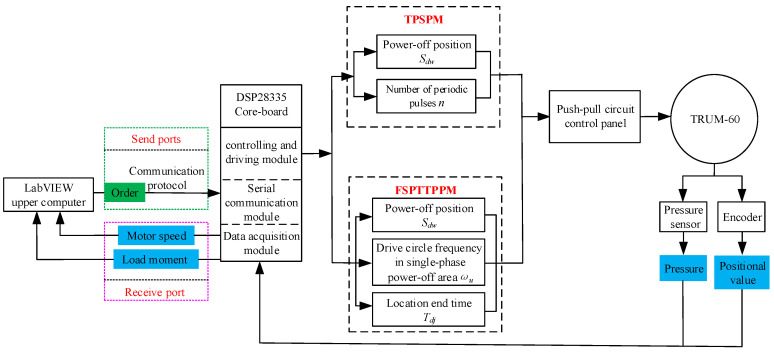
Block diagram of the motor position test system.

**Figure 16 micromachines-13-01542-f016:**
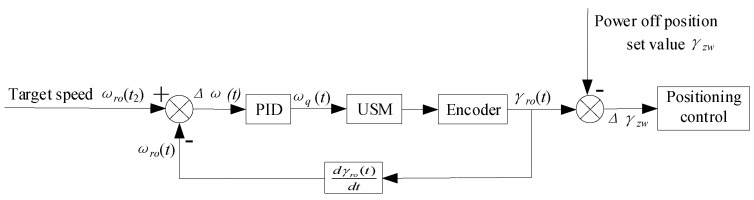
Block diagram of the control system for positioning control.

**Figure 17 micromachines-13-01542-f017:**
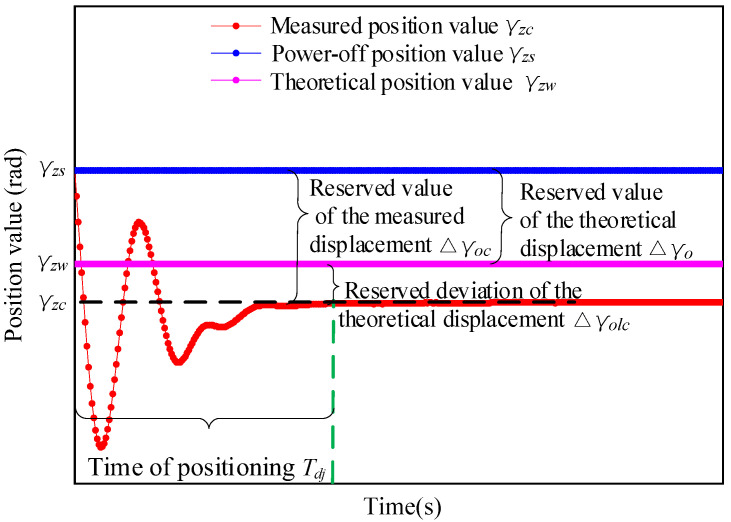
Plot showing the definitions of the different reserved values.

**Figure 18 micromachines-13-01542-f018:**
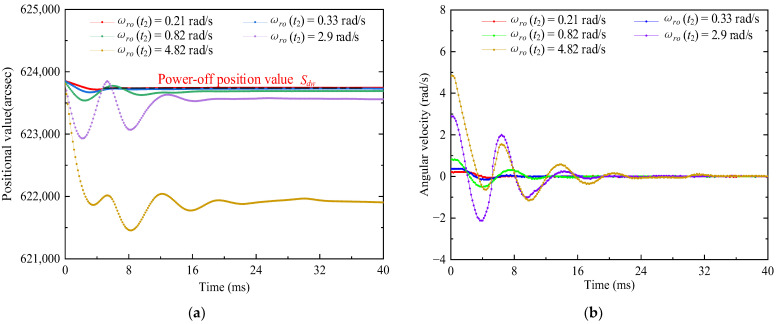
Measured values of the TPSPM (**a**) position and (**b**) angular rotation speed as a function of time at different initial angular velocities.

**Figure 19 micromachines-13-01542-f019:**
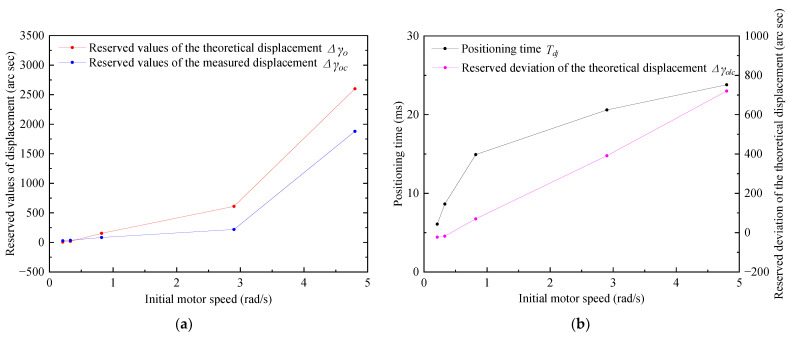
Changing trend of the displacement reservation value and positioning time. (**a**) Shows the trend of $\Delta \gamma_{o}$ and $\Delta \gamma_{oc}$, whereas (**b**) shows the trend of $T_{dj}$ and $\Delta \gamma_{olc}$ with increasing initial angular velocity.

**Figure 20 micromachines-13-01542-f020:**
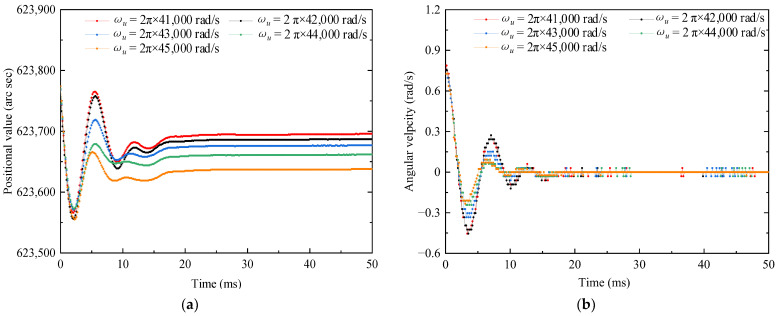
Measurement of the positioning of the single-phase power-off period in terms of the (**a**) measured position value and (**b**) angular velocity at different driving circle frequencies as a function of time.

**Figure 21 micromachines-13-01542-f021:**
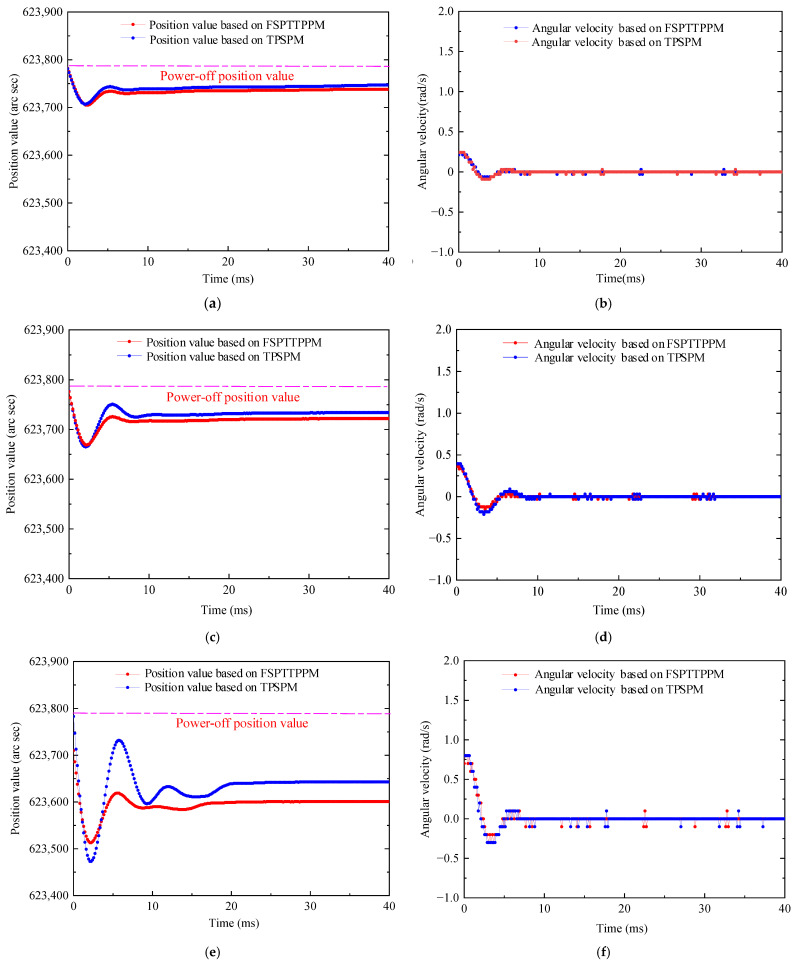
Measured position and angular velocity values obtained by applying the two positioning strategies investigated in this study. (**a**) Position curve for the initial angular velocity of 0.24 rad/s; (**b**) angular velocity curve for the initial angular velocity of 0.24 rad/s; (**c**) position curve for the initial angular velocity of 0.36 rad/s; (**d**) angular velocity curve for the initial angular velocity of 0.36 rad/s; (**e**) position curve for the initial angular velocity of 0.72 rad/s; (**f**) angular velocity curve for the initial angular velocity of 0.72 rad/s; (**g**) position curve for the initial angular velocity of 1.09 rad/s; (**h**) angular velocity curve for the initial angular velocity of 1.09 rad/s; (**i**) position curve for the initial angular velocity of 1.18 rad/s; (**j**) angular velocity curve for the initial angular velocity of 1.18 rad/s.

**Figure 22 micromachines-13-01542-f022:**
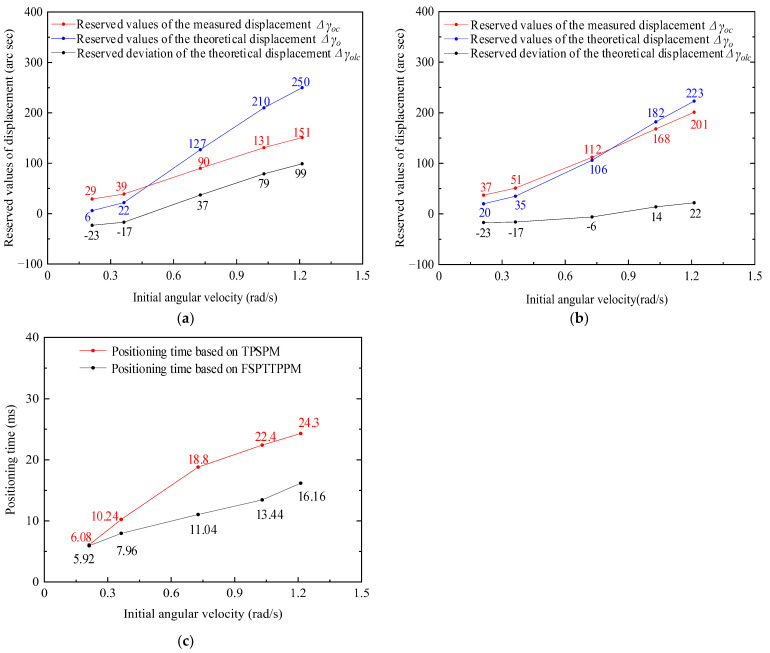
Analysis of the changing trend of (**a**) $\Delta \gamma_{oc}$, $\Delta \gamma_{o}$, and $\Delta \gamma_{olc}$ based on TPSPM, (**b**) $\Delta \gamma_{oc}$, $\Delta \gamma_{o}$, and $\Delta \gamma_{olc}$ based on FSPTTPPM, and (**c**) $T_{dj}$ based on the two positioning strategies.

**Figure 23 micromachines-13-01542-f023:**
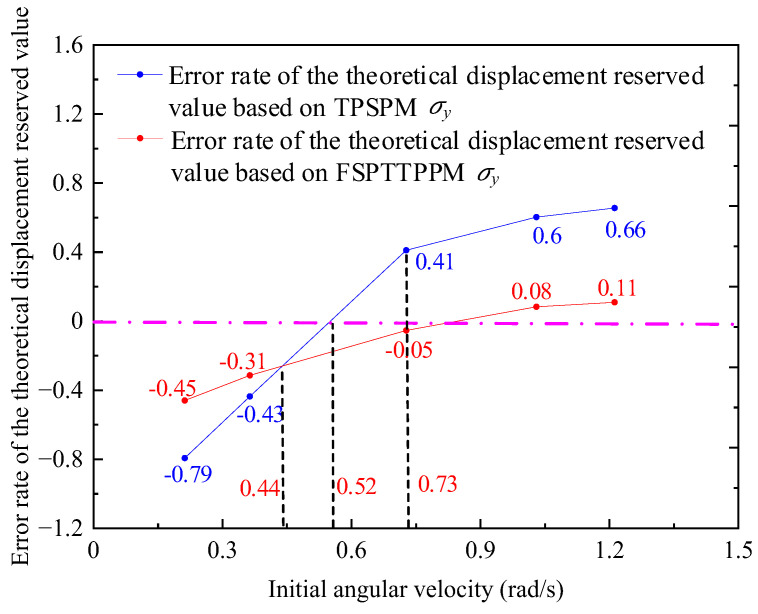
Trend of $\sigma_{y}$ and $T_{dj}$ of the two positioning strategies.

**Table 1 micromachines-13-01542-t001:** Parameter values.

Parameter	Description	Numerical Value (Unit)
$J_{ro}$	Moment of inertia of the shaft	1.7 × 10^−4^ (kg·m^2^)
$J_{st}$	Rotational inertia of the rotor	8 × 10^−8^ (kg·m^2^)
$\mu_{0}$	Sliding friction coefficient	0.3
$c_{st}$	Damping coefficient of the rotation direction of the rotor	0.05
$c_{ro}$	Damping coefficient of the shaft direction	0.04
$h_{ro}$	Length of the shaft	180 (mm)
$\mu_{ro}$	Poisson’s ratio of the shaft material	0.31
$E_{ro}$	Young’s modulus of the shaft material	20 (GPa)
R	Stator radius	30 (mm)
$D_{ro}$	Shaft diameter	50 (mm)
$k_{ro}$	Stiffness factor of the shaft direction	200 (GPa)

## Data Availability

The data presented in this study are available on request from the corresponding author. The data are not publicly available due to the experimental data in this paper being related to the aviation projects involving secrecy.

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
