# Peer review of "Research on Fast and Precise Positioning Strategy of an Ultrasonic Motor Based on the Ultrasonic Friction Reduction Theory"

_micromachines, 2022, doi:10.3390/mi13091542_

Round 1

Reviewer 1 Report

For the benefit of the reader, several points need clarifying, and certain statements require further justification. Following are some minor comments:

1.         According to Fig.23, the error rate of the theoretical displacement reserved value based on TPSPM is less than that of FSPTTPPM when the initial angular velocity is less than about 0.4 rpm. In fact, the initial angular velocity of most closed-loop positioning controller is quite slow, even lower than 0.1rad/s. It is suggested to explain that why the error rate of the theoretical displacement reserved value based on TPSPM is less than that of FSPTTPPM at low speed.

2.         In Fig. 7(b), the angular velocity of the particle Q is first accelerated and then decelerated by the correlation of the torque and the static friction torque of the shaft from ty to t4. It is interested that if the phenomenon can be replicated experimentally.

3.         According to Eq.(28),When TPSPM is used for positioning, the positioning time is the sum of  and . When FSPTTPPM is used for positioning, the positioning time is the sum of  and ,It is suggested to explain that why the location time based on TPSPM is less than that of FSPTTPPM.

4.         The eccentricity of the shaft and variable load mainly reduce the precision of the theoretical displacement reserved value. In this paper, this two factors are not considered further.

5.         The formatting error abound in this article. The font is not consistent. In 2.4.1, there exist serious typographical errors in the paragraph after Eq.(9).

Reviewer 2 Report

Dear Authors,

The manuscript should be revised before it is published. My comments and corrections are listed in the attachment.

Kind Regards 

Round 2

Reviewer 2 Report

Dear Authors,

You have adressed all my remarks and corrections in the revised version of the manuscript which can be published after minor revison (methodological and text editing mistakes like the one in line 649 "Mecha-tronics should be Mechatronics").

Kind Regards

Author Response

The word "Mecha-tronics" have been replaced with the word "Mechatronics", and marked with red font.